# Delayed repolarization and ventricular tachycardia in patients with heart failure and preserved ejection fraction

**Jae Hyung Cho[1], Derek Leong[1], Natasha Cuk[1], Joseph E. Ebinger[1], Catherine Bresee[2], Sung-Han Yoon[1], Ashkan Ehdaie[1], Michael Shehata[1], Xunzhang Wang[1], Sumeet S. Chugh[1], Eduardo Marbán[1], Eugenio Cingolani** [1]*

1 Smidt Heart Institute, Cedars-Sinai Medical Center, Los Angeles, California, United States of America,
2 Biostatistics and Bioinformatics Research Center, Cedars-Sinai Medical Center, Los Angeles, California, United States of America

* Eugenio.Cingolani@csmc.edu

**Data Availability Statement:** All relevant data are within the paper.

**Funding:** National Institutes of Health (RO1 HL135866 to EC and EM) and the Peer-Reviewed

## Abstract

Sudden death is the most common mode of mortality in patients with heart failure and preserved ejection fraction (HFpEF). Ventricular arrhythmias (VA) have been suspected as the etiology but the supporting evidence in patients with HFpEF is scarce. We sought to investigate VA prevalence, and to determine if VA are associated with prolonged repolarization, in patients with HFpEF. In a retrospective case-control study design, Cedars-Sinai patients who underwent prolonged ambulatory electrocardiographic monitoring (Zio Patch) between 2016 and 2018 were screened for a clinical diagnosis of HFpEF. Patients with normal diastolic and systolic function who underwent Zio Patch monitoring were also reviewed as controls. Multivariable logistic regression was used to compare the prevalence of rhythm disturbances in patients with and without HFpEF. Ventricular tachycardia (VT) was more prevalent in patients with HFpEF (37% vs. 16% in controls, p = 0.001). Most episodes were non-sustained except for one case of sustained VT in a patient with HFpEF. Covariate-adjusted logistic regression including HFpEF diagnosis, age, sex, body mass index, and the presence of comorbidities revealed that only HFpEF was associated with increased risk of VT (relative risk 2.86, p = 0.023). Subgroup-analyses revealed an association between increased QTc interval and risk of VT (460 ± 38 ms in HFpEF patients with VT vs. 445 ± 28 ms in HFpEF patients without VT, p = 0.03). Non-sustained VT was more prevalent in patients with HFpEF compared to patients without HFpEF, and QTc interval prolongation was associated with VT in HFpEF.

## Introduction

Heart failure and preserved ejection fraction (HFpEF) is increasing in incidence, rivaling heart failure and reduced ejection fraction (HFrEF) [1, 2]. Patients with HF have a poor prognosis, with a 75% mortality rate at 5 years, regardless of EF [3]. Unlike HFrEF, for which numerous medical and device therapies have been proven to reduce mortality, no treatment has been

Medical Research Program of the US Department
of Defense (PR150620 to EM). The sponsors
played no role in the the study design, data
collection and analysis, decision to publish, or
preparation of the manuscript.

**Competing interests:** The authors have declared
that no competing interests exist.

proven to prolong survival of patients with HFpEF. Specifically, traditional HF medications
such as beta-blockers, angiotensin-converting enzyme inhibitors, angiotensin receptor II
blockers and aldosterone antagonists, have failed to decrease mortality of patients with
HFpEF, although there were regional variations with aldosterone antagonist [4–7]. Although
sudden death is the most common mode of mortality in patients with HFpEF, the underlying
mechanisms remain unclear [8]. Ventricular arrhythmias (VA) may play a role [9]; however,
this remains untested in patients. Preclinical studies have revealed that VA are common and
associated with sudden death in a rat HFpEF model [10, 11], with underlying repolarization
delays revealed by electrocardiogram (ECG), optical mapping and patch clamp [12]. The pres-
ent study sought to investigate VA prevalence, and to determine if VA are associated with pro-
longed repolarization, in patients with HFpEF.

## Materials and methods

### Study approval

This study was presented to the Institutional Review Board of Cedars-Sinai Medical Center
and approval was obtained before the initiation of data collection. The nature of this study is
medical record review of patients with HFpEF compared to patients without HFpEF.

### Patch ambulatory monitoring of ECG (Zio Patch)

We identified all patients (inpatients and outpatients) who underwent ambulatory monitoring
of ECG by Zio Patch (iRhythm, San Francisco, CA) for any reason from January 2016 to
December 2018 (3 years, N = 2,913, Fig 1). Our institution currently uses Zio Patch for ambu-
latory ECG monitoring up to 14 days.

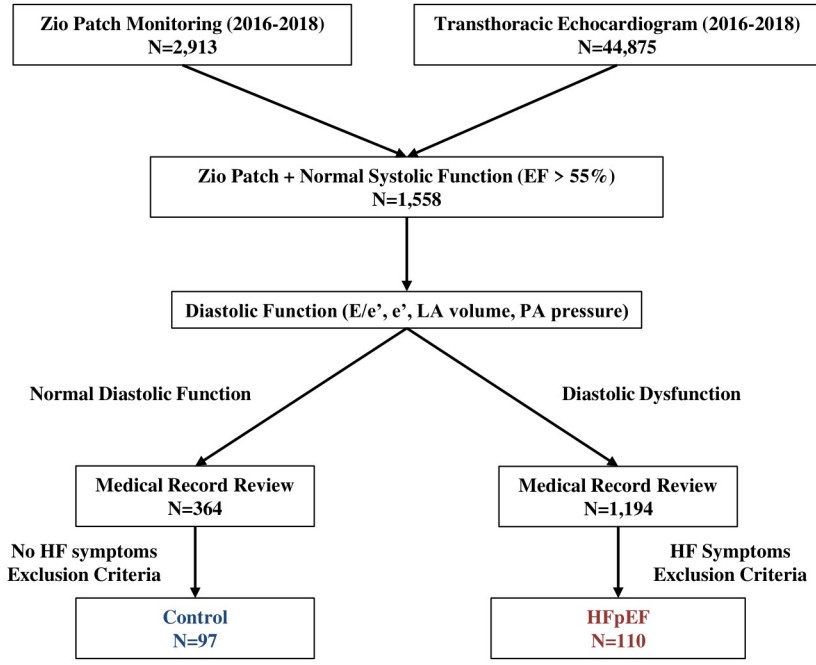

**Fig 1. Patient identification algorithm.**

### Identification of patients with diastolic dysfunction

The Cedars-Sinai Echocardiography Database was queried for all transthoracic echocardiograms completed on patients between January 2016 to December 2018 (N = 44,875, Fig 1). Patients with normal systolic function (EF > 55%) who underwent Zio Patch monitoring were identified (N = 1,558, Fig 1). Echocardiograms were grouped as demonstrating either normal or abnormal diastolic function based on guideline criteria from the American Society of Echocardiography (average E/e' > 14, septal e' velocity < 7 cm/s or lateral e' velocity < 10 cm/s, tricuspid regurgitation velocity > 2.8 m/s and left atrial volume index > 34 ml/m$^2$) [13]. The severity of diastolic dysfunction (grade I, II, and III) was also determined based on guidelines [13].

### Identification of patients with HFpEF vs. without HFpEF

Medical records of patients with diastolic dysfunction of any severity (N = 1,194) were reviewed to identify those with a clinical diagnosis of HFpEF (HF with symptoms, normal EF and diastolic dysfunction on transthoracic echocardiogram). Patients with diastolic dysfunction (due to old age) without documented HF symptoms were excluded. Patients with the following conditions were also excluded: uncorrected primary left sided valvular heart disease (aortic stenosis/regurgitation or mitral stenosis/regurgitation), isolated right ventricular failure (pulmonary hypertension, pulmonary or tricuspid valvular disease, arrhythmogenic right ventricular cardiomyopathy and congenital heart disease), pericardial disease (cardiac tamponade and constrictive pericarditis) and specific cardiomyopathies (amyloidosis, sarcoidosis, hypertrophic cardiomyopathy and restrictive cardiomyopathy) [1]. Patients who had normal diastolic and systolic function (N = 364) were also reviewed and only those without HF symptoms were selected to serve as controls for comparison.

### Data collection

Chart reviews were performed (authors J.C. and D.L.) to collect relevant data. Baseline characteristics included age, sex, body mass index, hypertension, diabetes mellitus, hyperlipidemia, coronary artery disease, chronic kidney disease, atrial fibrillation, EF and diastolic function from the echocardiogram, medications (beta-blocker, calcium channel blocker, amiodarone and QT prolonging medications), ECG parameters (heart rate [HR], PR interval, QRS width, QT interval and QTc interval) and indications for Zio Patch monitoring (syncope, stroke, atrial fibrillation, palpitations, bradycardia, dizziness, etc.). Zio Patch results were reviewed to identify the prevalence of rhythm disturbances such as ventricular tachycardia (VT), supraventricular tachycardia, atrial fibrillation, atrial flutter, sinus pause and atrioventricular block.

### Statistical analyses

SPSS was used to perform the statistical analysis. Baseline patient characteristics are presented as numbers and percentages for categorical variables and mean ± standard deviation for continuous variables. Comparisons of categorical variables were performed using Pearson's chi-square and comparisons of continuous variables were performed using independent t-test. Both simple and covariate-adjusted multiple logistic regression modeling was used to compare the likelihood of VT between patients with and without HFpEF. Results were considered significant at $p < 0.05$.

## Results

### Patients with HFpEF vs. without HFpEF

A total of 110 patients with HFpEF underwent Zio Patch monitoring during the study period (Fig 1). As controls, 97 patients with normal diastolic and systolic function and Zio Patch

monitoring were identified during the same time period. Most of the patients underwent Zio Patch within a week from echocardiogram (only 2 patients in each group underwent Zio Patch after 30 days of echocardiogram).

## Baseline characteristics

Baseline characteristics of the HFpEF and control groups are shown in Table 1. Patients with HFpEF were older and had increased prevalence of hypertension, hyperlipidemia, coronary artery disease, chronic kidney disease and atrial fibrillation than patients without HFpEF. Male to female ratio of patients were not significantly different between the two groups, nor was the prevalence of diabetes mellitus or difference in body mass index. The severity of diastolic dysfunction in the HFpEF group was grade I in 64%, and grade II in 36%, of cases. None of the HFpEF patients met criteria for grade III diastolic dysfunction. Patients with HFpEF were taking more beta-blockers, calcium channel blockers and amiodarone. Approximately a quarter of patients in each group were taking QT prolonging medications (including amiodarone) at the time of Zio Patch monitoring. Baseline heart rates were not significantly different between the two groups. PR interval and QTc interval were more prolonged in patients with HFpEF compared to controls. QRS width was not different between control and HFpEF patients. Indications for Zio Patch monitoring were similar between the two groups.

## Prevalence of rhythmic disturbances

VT was more prevalent in patients with HFpEF compared to controls (41/110 = 37% in HFpEF vs. 16/97 = 16% in controls, p = 0.001) (Table 2). Most of the VT episodes were non-sustained (less than 30 seconds by definition) except for one episode of sustained VT in a HFpEF patient. The average number of VT beats and durations were higher in patients with HFpEF than controls, but did not reach statistical significance (average number of beats 12.1 ± 16.1 vs. 8.3 ± 7.8 in controls, p = 0.237 and average duration 5.8 ± 7.5 seconds vs. 3.9 ± 2.5 seconds in controls, p = 0.182). Supraventricular tachycardia was slightly more prevalent in patients with HFpEF. Prevalence of atrial fibrillation and flutter, and atrioventricular block, were not statistically different. Sinus pause was slightly more prevalent in HFpEF patients.

## Simple logistic regression

Only HFpEF (relative risk [RR] 3.00, 95% confidence interval [CI] 1.55–5.83, p = 0.001) and age (RR 1.05, 95% CI 1.02–1.08, p = 0.001) was found to be associated with increased risk of VT by simple logistic regression (Fig 2). Other risk factors (sex, body mass index, hypertension, diabetes mellitus, hyperlipidemia, coronary artery disease, chronic kidney disease and atrial fibrillation) were not found to be independently associated with increased risk of VT.

## Covariate-adjusted logistic regression

After covarying for all factors, only HFpEF was associated with increased risk of VT compared with controls (RR 2.86, 95% CI 1.15–7.08, p = 0.023) (Fig 2). Older age demonstrated a trend towards increased VT risk, but did not reach statistical significance (RR 1.03, 95% CI 0.99–1.07, p = 0.068). Other risk factors (sex, body mass index, hypertension, diabetes mellitus, hyperlipidemia, coronary artery disease, chronic kidney disease and atrial fibrillation) were not found to be associated with VT risk.

**Table 1. Baseline characteristics.**

|  | Control (N = 97) | HFpEF (N = 110) | P value |
|---|---|---|---|
| Age (mean ± SD) | 67 ± 9 | 80 ± 12 | < 0.001 |
| Sex (male) | 37 (38%) | 52 (47%) | 0.247 |
| Body mass index (mean ± SD) | 27 ± 6 | 27 ± 6 | 0.982 |
| Hypertension | 60 (62%) | 96 (87%) | < 0.001 |
| Diabetes mellitus | 15 (15%) | 23 (21%) | 0.313 |
| Hyperlipidemia | 56 (58%) | 93 (85%) | < 0.001 |
| Coronary artery disease | 15 (15%) | 60 (55%) | < 0.001 |
| Chronic kidney disease | 4 (4%) | 28 (25%) | < 0.001 |
| Atrial fibrillation | 21 (22%) | 44 (40%) | 0.005 |
| Ejection fraction (mean ± SD) | 64 ± 6 | 65 ± 8 | 0.251 |
| Diastolic function (mean ± SD) |  |  |  |
| Normal | 97 (100%) | 0 (0%) | < 0.001 |
| Grade I diastolic dysfunction | 0 (0%) | 70 (64%) |  |
| Grade II diastolic dysfunction | 0 (0%) | 40 (36%) |  |
| Grade III diastolic dysfunction | 0 (0%) | 0 (0%) |  |
| MV inflow E wave (cm/s) | 79.8 ± 15.9 | 85.6 ± 28.6 | 0.077 |
| MV inflow A wave (cm/s) | 64.4 ± 14.8 | 93.3 ± 30.0 | < 0.001 |
| E/A ratio | 1.3 ± 0.2 | 1.0 ± 0.6 | < 0.001 |
| Lateral e' wave | 10.2 ± 2.5 | 7.3 ± 2.2 | < 0.001 |
| Lateral E/e' | 8.5 ± 2.5 | 12.9 ± 6.5 | < 0.001 |
| TR velocity (m/s) | 2.2 ± 0.4 | 2.6 ± 0.5 | < 0.001 |
| LA volume index (ml/m$^2$) | 26.4 ± 13.0 | 32.8 ± 13.6 | 0.004 |
| Heart failure symptoms | 0 (0%) | 110 (100%) | < 0.001 |
| Laboratory data (mean ± SD) |  |  |  |
| BNP (pg/ml) | 93.6 ± 110.0 | 351.4 ± 464.9 | 0.008 |
| Na (mmol/L) | 140.2 ± 4.0 | 139.7 ± 3.5 | 0.427 |
| K (mmol/L) | 4.1 ± 0.4 | 4.2 ± 0.6 | 0.082 |
| Ca (mg/dL) | 9.0 ± 0.6 | 9.1 ± 0.5 | 0.285 |
| Creatinine (mg/dL) | 0.9 ± 0.5 | 1.4 ± 1.6 | 0.009 |
| Medications |  |  |  |
| Beta-blocker | 27 (28%) | 55 (50%) | 0.001 |
| Calcium channel blocker | 16 (16%) | 39 (35%) | 0.002 |
| Amiodarone | 2 (2%) | 16 (15%) | 0.001 |
| QT prolonging medication | 27 (28%) | 27 (25%) | 0.591 |
| ECG parameters |  |  |  |
| HR (bpm, mean ± SD) | 68 ± 13 | 69 ± 15 | 0.412 |
| PR interval (ms, mean ± SD) | 165 ± 37 | 192 ± 50 | < 0.001 |
| QRS width (ms, mean ± SD) | 93 ± 17 | 100 ± 33 | 0.051 |
| QT interval (ms, mean ± SD) | 414 ± 39 | 426 ± 48 | 0.043 |
| QTc interval (ms, mean ± SD) | 432 ± 29 | 451 ± 33 | < 0.001 |
| Indications for ECG monitoring |  |  |  |
| Syncope | 23 (24%) | 28 (25%) | 0.771 |
| Stroke | 24 (25%) | 19 (17%) | 0.186 |
| Atrial fibrillation | 15 (15%) | 21 (19%) | 0.492 |
| Palpitation | 20 (21%) | 10 (9%) | 0.018 |
| Bradycardia | 6 (6%) | 14 (13%) | 0.112 |
| Dizziness | 5 (5%) | 1 (1%) | 0.069 |
| Others | 4 (4%) | 17 (15%) | 0.010 |

**Table 2. Prevalence of rhythm disturbances.**

|  | Control (N = 97) | HFpEF (N = 110) | P value |
|---|---|---|---|
| Ventricular tachycardia | 16 (16%) | 41 (37%) | 0.001 |
| Sustained | 0 (0%) | 1 (1%) | 0.347 |
| Non-sustained | 16 (16%) | 41 (37%) | 0.001 |
| Supraventricular tachycardia | 70 (72%) | 92 (84%) | 0.046 |
| Atrial fibrillation | 9 (9%) | 12 (11%) | 0.699 |
| Atrial flutter | 1 (1%) | 3 (3%) | 0.379 |
| Sinus pause | 0 (0%) | 5 (5%) | 0.034 |
| Atrioventricular block | 0 (0%) | 1 (1%) | 0.235 |

## Electrocardiographic parameter comparison

We investigated whether any of the electrocardiographic parameters are associated with increased risk of VT in patients with HFpEF (Table 3). HR, PR interval, QRS width and QT interval were not different between HFpEF patients with and without VT. Only QTc interval was more prolonged in HFpEF patients with VT compared to HFpEF patients without VT (460 ± 38 ms vs. 445 ± 28 ms, p = 0.032).

## Discussion

In this retrospective case control study, non-sustained VT was more prevalent in patients with HFpEF compared to patients without HFpEF. QTc interval was prolonged in patients with HFpEF compared to controls, and the prolongation was more prominent in HFpEF patients with VT compared to HFpEF patients without VT (Fig 3).

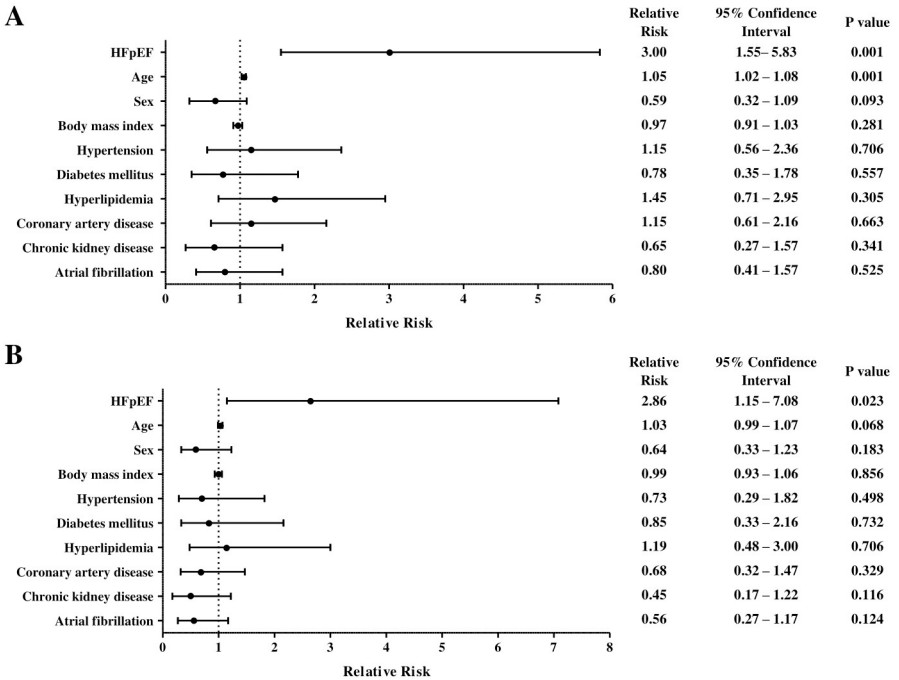

**Fig 2. Simple (A) and multiple (B) logistic regression of the likelihood of ventricular tachycardia.**

**Table 3. Subgroup-analyses of HFpEF patients.**

|  | VT (N = 41) | No VT (N = 69) | P value |
|---|---|---|---|
| Age (mean ± SD) | 82 ± 8 | 80 ± 14 | 0.265 |
| Sex (male) | 23 (56%) | 29 (42%) | 0.260 |
| Body mass index (mean ± SD) | 26 ± 5 | 27 ± 7 | 0.301 |
| Hypertension | 34 (83%) | 62 (90%) | 0.296 |
| Diabetes | 6 (15%) | 17 (25%) | 0.216 |
| Hyperlipidemia | 33 (80%) | 60 (87%) | 0.390 |
| Coronary artery disease | 21 (51%) | 39 (57%) | 0.595 |
| Chronic kidney disease | 7 (17%) | 21 (30%) | 0.122 |
| Atrial fibrillation | 12 (29%) | 32 (46%) | 0.078 |
| Ejection fraction (mean ± SD) | 64 ± 8 | 66 ± 8 | 0.113 |
| Diastolic function |  |  |  |
| Grade I diastolic dysfunction | 28 (68%) | 42 (61%) | 0.612 |
| Grade II diastolic dysfunction | 13 (32%) | 27 (39%) |  |
| Grade III diastolic dysfunction | 0 (0%) | 0 (0%) |  |
| Medications |  |  |  |
| Beta-blocker | 14 (34%) | 41 (59%) | 0.010 |
| Calcium channel blocker | 15 (37%) | 24 (35%) | 0.851 |
| Amiodarone | 3 (7%) | 13 (19%) | 0.099 |
| QT prolonging medication | 8 (20%) | 19 (28%) | 0.335 |
| ECG parameters |  |  |  |
| HR (bpm, mean ± SD) | 69 ± 13 | 70 ± 16 | 0.805 |
| PR interval (ms, mean ± SD) | 183 ± 44 | 198 ± 53 | 0.122 |
| QRS width (ms, mean ± SD) | 99 ± 22 | 101 ± 38 | 0.753 |
| QT interval (ms, mean ± SD) | 435 ± 49 | 421 ± 47 | 0.138 |
| QTc interval (ms, mean ± SD) | 460 ± 38 | 445 ±28 | 0.032 |

In prospective analyses of rhythm in the Dahl salt-sensitive rat model of HFpEF, VA are more frequent in HFpEF rats than in controls, and these VA are associated with sudden death [11, 12]. The underlying mechanisms include repolarization delays, conduction slowing and inhomogeneities of excitation. Delayed repolarization was manifest in ECG as prolonged QT/QTc interval, and in optical mapping and patch clamp as action potential prolongation [12]. HFpEF rats showed multiple re-entry circuits in optical mapping and increased fibrosis in tissue sections [12]. These anatomical and functional re-entry circuits both contribute to the increased propensity to VA. Ambulatory ECG monitoring revealed that VA were the cause of 75% of documented sudden deaths in HFpEF rats [11]. Here we have described, in human patients, associations among HFpEF, VT prevalence, and QTc interval prolongation. Our clinical data are consistent with the notion that inhomogeneities of excitation and repolarization underlie non-sustained VT in patients with HFpEF, but the insights here fall far short of establishing causality; instead, they should be considered hypothesis-generating.

Sudden death is the leading mode of mortality in patients with HFpEF; however, the mechanisms of sudden death have not been studied in this population. The World Health Organization defines sudden death as death occurring less than 1 hour from acute changes in witnessed cases or found dead within 24 hours in unwitnessed cases [14]. Sudden death can be divided into sudden cardiac death (SCD) and sudden non-cardiac death (SNCD) based on the etiology. In studies to date, most SCD are due to VA, which can be precipitated in coronary artery disease, and cardiomyopathy [15]. SNCD includes causes of death such as pulmonary disease

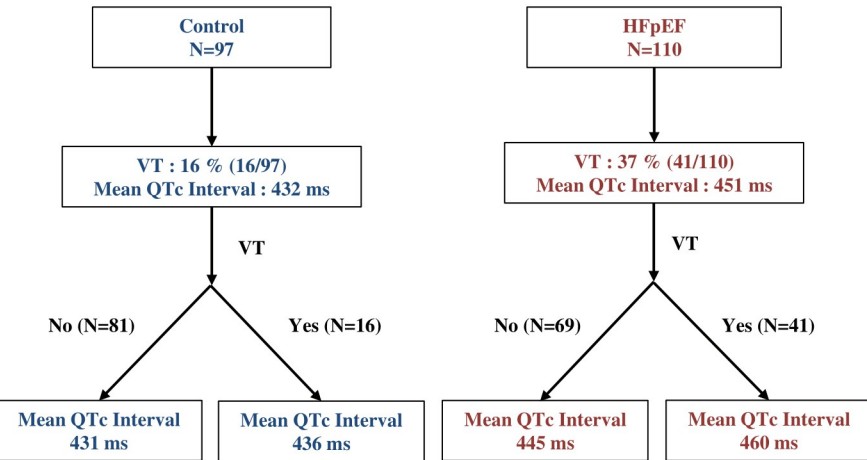

**Fig 3. Prolonged QTc interval in HFpEF patients compared to controls.**

(40%), infectious disease (20%), cerebrovascular disease (18%) and neurologic diseases (8%) [14]. The investigation of sudden death is challenging due to its inherent rapidity, and by its unpredictability [16].

Non-sustained VT (less than 30 seconds) has been recorded in a variety of patients, from healthy individuals to patients with significant heart disease. The actionability of non-sustained VT is highly debatable, but it generally portends enhanced risk. For example, in patients with non-ST elevation acute coronary syndrome, non-sustained VT after 48 hours of admission is associated with more than a 2-fold increase in sudden death [17], but using an automatic external defibrillator does not decrease mortality [18]. In patients with HFrEF (ischemic or nonischemic), non-sustained VT is present in 30–80% and is an independent marker of overall mortality and sudden death [19, 20], but this criterion is not used in the decision to place an implantable cardioverter-defibrillator (ICD). In contrast, in cases of hypertrophic cardiomyopathy, ICD placement is generally indicated for patients with non-sustained VT (more than 3 beats and greater than 120 bpm) [21]. The prevalence and importance of non-sustained VT have not been studied in patients with HFpEF. Our findings suggest that non-sustained VT may be more frequent in patients with HFpEF, but this prediction, and its prognostic implications, should be tested in large-scale prospective studies.

The conclusions must be tempered by several limitations. First, the observational nature of this study introduces selection bias related to indications for both a Zio Patch and an echocardiogram. The indications for Zio Patch monitoring and echocardiography, however, were not different between HFpEF patients and controls, indicating at least similar reasons for referrals. Second, the study design allows us to determine associations between VT and HFpEF; however, mechanistic and causal links remain to be elucidated. Further, the impact of VT on sudden death was not investigated here. Third, although the initial pool of patients is large, the actual dataset used for analysis is small. This is mainly due to the limited use of Zio Patch monitoring. Larger-scale prospective community surveillance is required for more definitive conclusions [15]. Finally, we have not investigated whether prolonged QTc interval is a cause of VT or just an associated phenomenon. Our pre-clinical findings showed that prolongation of QTc interval exacerbated VT, and shortening of QTc interval diminished susceptibility to VT [12, 22]. Once again, large prospective studies are required to test the relationship between QTc interval and VT in HFpEF.

## Author Contributions

**Conceptualization:** Jae Hyung Cho, Derek Leong, Joseph E. Ebinger, Eugenio Cingolani.

**Data curation:** Jae Hyung Cho, Derek Leong, Natasha Cuk, Joseph E. Ebinger.

**Formal analysis:** Jae Hyung Cho, Catherine Bresee, Eugenio Cingolani.

**Investigation:** Jae Hyung Cho, Derek Leong, Natasha Cuk, Joseph E. Ebinger, Sung-Han Yoon, Ashkan Ehdaie, Michael Shehata, Xunzhang Wang, Sumeet S. Chugh, Eduardo Marbán, Eugenio Cingolani.

**Methodology:** Jae Hyung Cho, Derek Leong, Natasha Cuk, Joseph E. Ebinger, Catherine Bresee, Sung-Han Yoon, Ashkan Ehdaie, Michael Shehata, Xunzhang Wang, Sumeet S. Chugh, Eugenio Cingolani.

**Supervision:** Ashkan Ehdaie, Michael Shehata, Xunzhang Wang, Sumeet S. Chugh, Eduardo Marbán, Eugenio Cingolani.

**Writing – original draft:** Jae Hyung Cho.

**Writing – review & editing:** Sumeet S. Chugh, Eduardo Marbán, Eugenio Cingolani.

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
