## [Decision Letter · Decision Letter 0]

10 Dec 2020

PONE-D-20-33005

Delayed Repolarization and Ventricular Tachycardia

in Patients with Heart Failure and Preserved Ejection Fraction

PLOS ONE

Dear Dr. Cingolani,

Thank you for submitting your manuscript to PLOS ONE. After careful consideration, we feel that it has merit but does not fully meet PLOS ONE’s publication criteria as it currently stands. Therefore, we invite you to submit a revised version of the manuscript that addresses the points raised during the review process.

Please address comments indicated by the Reviewers.

We look forward to receiving your revised manuscript.

Kind regards,

Elena G. Tolkacheva, PhD

Academic Editor

PLOS ONE

Journal Requirements:

2. Please include your tables as part of your main manuscript and remove the individual files. Please note that supplementary tables should be uploaded as separate "supporting information" files.

- https://www.sciencedirect.com/science/article/abs/pii/S0735109720310627?via%3Dihub

The text that needs to be addressed involves the majority of the abstract.

In your revision ensure you cite all your sources (including your own works), and quote or rephrase any duplicated text outside the methods section. Further consideration is dependent on these concerns being addressed.

Reviewers' comments:

Reviewer's Responses to Questions

**Comments to the Author**

1. Is the manuscript technically sound, and do the data support the conclusions?

Reviewer #1: Partly

Reviewer #2: Partly

2. Has the statistical analysis been performed appropriately and rigorously? 

Reviewer #1: Yes

Reviewer #2: Yes

3. Have the authors made all data underlying the findings in their manuscript fully available?

Reviewer #1: No

Reviewer #2: No

4. Is the manuscript presented in an intelligible fashion and written in standard English?

Reviewer #1: Yes

Reviewer #2: Yes

5. Review Comments to the Author

Reviewer #1: The authors of “Delayed repolarization and ventricular tachycardia in patients with Heart Failure Preserve Ejection fraction” attempts to explore the association between HFpEF and ventricular arrhythmias. The hypothesis presented is relevant and statistically methods appear sound. Caution, however, needs to be made in some of the conclusions drawn from this rather small retrospective dataset, among some other minor comments.

1. Caution concluding the enhance propensity of VT contributes to sudden death in these patients as there was no long term follow up or outcome data presented. As mentioned by the authors “non-sustained VT has been recorded in the variety of patients, from healthy individual to patient with significant heart disease. The actionability of non-sustained VT is high debateable …..” I would encourage the authors to focus on the increase prevalence of VT demonstrated by the data presented, underlying mechanisms and be cautious in suggesting a direct association with NSVT and SCD.

2. In the statistical methods the authors have stated that results were considered significant at p < 0.05, please avoid using qualifying words through the manuscript such as “modestly” or “strong trend” to imply statistical significance.

3. In the methods section please clarify if the patients included were inpatients, ambulatory care patients or both.

4. Considering echocardiographic data was used to define HFpEF, these data should be provided.

5. Were laboratory values available around the time of echocardiography, specifically NT-proBNP or metabolic panel?

6. If these data suggest an enhance propensity to VT in HFpEF, why does the frequency of arrhythmias, specifically VT, not increase with severity of diastolic dysfunction.

7. If patients in the HFpEF were taking more rate controlling medication, is there an explanation for why was there no difference in heart rates between to two groups?

8. While the initial pool of patients is large, the actually dataset used for analysis is small and this should be specifically mentioned in the limitations.

Reviewer #2: This study examined Zio Patch data in patients with a clinical diagnosis of HFpEF. Patients with normal diastolic and systolic function served as controls. The results showed that VT was more prevalent in patients with HFpEF (versus those without this diagnosis).

Comments:

-How much time was there between the Zio Patch and TTE?

-The authors used the ASE guidelines to adjudicate evidence of diastolic dysfunction however there is significant controversy on how age influences diastolic parameters – namely septal and lateral e’ velocities. An older patient with lower e’ velocities may also have higher E/e’ ratios. These patients might be categorized with diastolic dysfunction when this may not be the case.

-How were the medical records used to determine the presence of clinical HFpEF? Was BNP used? What criteria were used to classify patients as HFpEF. Relying on the coded medical chart alone leaves a lot of opportunity to introduce errors.

-Rather than say mild, moderate, severe diastolic dysfunction, use grade I, II, or III if this is what the authors are suggesting. I am also surprised that no patients had grade III diastolic dysfunction using criteria E/A>2.

-It’s odd that patients with other risk factors did not have higher chance of VT (namely CAD). This might be due to a sample size issue. Could the authors expand the date range if Zio Patch data was available before 2016.

-I’m sure a number of these patients underwent cardiac MRI…of the ones who underwent CMR, what fraction had myocardial LGE present in this study and how does this impact the data.

-The wording “mode of exodus” in the abstract , introduction, and discusssion is odd…perhaps change to “most common cause of mortality”

6. PLOS authors have the option to publish the peer review history of their article (what does this mean?). If published, this will include your full peer review and any attached files.

Reviewer #1: No

Reviewer #2: No

---

## [Author Response · Author response to Decision Letter 0]

3 Feb 2021

We have revised our manuscript following the two reviewers' comments.

---

## [Decision Letter · Decision Letter 1]

1 Jul 2021

Delayed repolarization and ventricular tachycardia

in patients with heart failure and preserved ejection fraction

PONE-D-20-33005R1

Dear Dr. Cingolani,

We’re pleased to inform you that your manuscript has been judged scientifically suitable for publication and will be formally accepted for publication once it meets all outstanding technical requirements.

Kind regards,

Elena G. Tolkacheva, PhD

Academic Editor

PLOS ONE

Additional Editor Comments (optional):

Reviewers' comments:

Reviewer's Responses to Questions

**Comments to the Author**

1. If the authors have adequately addressed your comments raised in a previous round of review and you feel that this manuscript is now acceptable for publication, you may indicate that here to bypass the “Comments to the Author” section, enter your conflict of interest statement in the “Confidential to Editor” section, and submit your "Accept" recommendation.

Reviewer #1: All comments have been addressed

Reviewer #2: All comments have been addressed

2. Is the manuscript technically sound, and do the data support the conclusions?

Reviewer #1: Yes

Reviewer #2: Yes

3. Has the statistical analysis been performed appropriately and rigorously? 

Reviewer #1: Yes

Reviewer #2: Yes

4. Have the authors made all data underlying the findings in their manuscript fully available?

Reviewer #1: No

Reviewer #2: Yes

5. Is the manuscript presented in an intelligible fashion and written in standard English?

Reviewer #1: Yes

Reviewer #2: Yes

6. Review Comments to the Author

Reviewer #1: Thank you for addressing concerns raised and adding requested data (echocardiographic data and relevant laboratory studies).

Reviewer #2: This is a revision of the manuscript examining the association of ventricular arrhythmias and HFpEF patients. I have reviewed the responses the authors have submitted. They have addressed them as best as possible. It is surprising that so few patients underwent CMR as often the presence of scar can be linked to ventricular arrhythmias and thus help more on the mechanistic aspect of this study.

7. PLOS authors have the option to publish the peer review history of their article (what does this mean?). If published, this will include your full peer review and any attached files.

Reviewer #1: No

Reviewer #2: No

---

## [Editor Report · Acceptance letter]

5 Jul 2021

PONE-D-20-33005R1 

Delayed repolarization and ventricular tachycardia in patients with heart failure and preserved ejection fraction 

Dear Dr. Cingolani:

I'm pleased to inform you that your manuscript has been deemed suitable for publication in PLOS ONE. Congratulations! Your manuscript is now with our production department. 

Kind regards, 

on behalf of

Dr. Elena G. Tolkacheva 

Academic Editor

PLOS ONE